



# Source Apportionment of Soot Particles and Aqueous-Phase Processing of Black Carbon Coatings in an Urban Environment

Ryan N. Farley[1,2], Sonya Collier[1,a], Christopher D. Cappa[3], Leah R. Williams[4], Timothy B. Onasch[4], Lynn M. Russell[5], Hwajin Kim[1,b], Qi Zhang[1,2]

[1]Department of Environmental Toxicology, University of California Davis, CA, 95616, USA
[2]Agricultural and Environmental Chemistry Graduate Group, University of California Davis, CA 95616, USA
[3]Department of Civil and Environmental Engineering, University of California Davis, CA , 95616, USA
[4]Aerodyne Research Inc., Billerica, MA, 01821, USA
[5]Scripps Institution of Oceanography, University of California San Diego, CA, 92037, USA
[a]Now at: California Air Resources Board, 1001 I Street, Sacramento, CA 95814, USA
[b]Now at: Seoul National University, Seoul, South Korea

*Correspondence to:* Qi Zhang (dkwzhang@ucdavis.edu)

**Abstract**

15        The impacts of soot particles on climate and human health depend on the concentration of black carbon (BC) as well as the thickness and composition of the coating material, i.e., organic and inorganic compounds internally mixed with BC. In this study, the size-resolved chemical composition of BC-containing aerosol was measured using a high-resolution soot-particle aerosol mass spectrometer (SP-AMS) during wintertime in Fresno, California, a location influenced by abundant combustion emissions and frequent fog events. Concurrently, particle optical properties were measured to investigate the BC

absorption enhancement. Positive matrix factorization (PMF) analysis was performed on the SP-AMS mass spectral measurements to explore the sources of soot particles and the atmospheric processes affecting the properties of BC coatings. The analysis revealed that residential wood burning and traffic are the dominant sources of soot particles. Alongside primary soot particles originating from biomass burning (BBOA$_{BC}$) and vehicles (HOA$_{BC}$) two distinct types of processed BC-containing aerosol were identified: fog-related oxidized organic aerosol (FOOA$_{BC}$) and winter-background OOA$_{BC}$

(WOOA$_{BC}$). Both types of OOA$_{BC}$ showed evidence of having undergone aqueous processing, albeit with differences. The concentration of FOOA$_{BC}$ was substantially elevated during fog events, indicating the formation of aqueous secondary organic aerosol (aqSOA) within fog droplets. On the other hand, WOOA$_{BC}$ was present at a relatively consistent concentration throughout the winter and is likely related to the formation of secondary organic aerosol (SOA) in both the gas phase and aerosol liquid water. By comparing the chemical properties and temporal variations of FOOA$_{BC}$ and WOOA$_{BC}$, we gain

insights into the key aging processes of BC aerosol. It was found that aqueous-phase reactions facilitated by fog droplets had a significant impact on the thickness and chemical composition of BC coatings, thereby affecting the light absorption and hygroscopic properties of soot particles. These findings underscore the important role of chemical reactions occurring within clouds and fogs and influencing the climate forcing of BC aerosol in the atmosphere.



**1.  Introduction**

Soot particles, also known as black carbon (BC) aerosol, are produced during the incomplete combustion of biomass and fossil fuels. BC strongly absorbs solar radiation and significantly influences regional and global climate. Indeed, it is considered the second largest global warming agent after $CO_2$ (IPCC, 2021; Kumar et al., 2018; Ramanathan and Carmichael, 2008). In addition to direct effects on radiative forcing, BC aerosol can also alter cloud properties by acting as cloud
condensation nuclei (CCN), increase cloud evaporation rates, and impede atmospheric mixing (Bond and Bergstrom, 2006; Koch and Del Genio, 2010; Petäjä et al., 2016).

Soot particles are often emitted as highly fractal structures, but can become internally mixed with secondary organic aerosol (SOA) and inorganic species through condensation and coagulation processes (Bhandari et al., 2019; Zhang et al., 2008), resulting in compaction into a spherical shape (Lee et al., 2019). The mixing state and coating composition of BC have
important implications for the optical properties and climatic impacts of soot aerosol. Although BC itself is hydrophobic, the mixing with hydrophilic material can convert soot particles into effective CCN, thereby promoting cloud formation and increasing the BC wet deposition rate (Wu et al., 2019). Furthermore, the presence of coating material can enhance the light absorption of BC through the so-called lensing effect (Cappa et al., 2012; Peng et al., 2016). The magnitude of this enhancement is dependent on the coating composition, with absorptive material such as brown carbon (BrC) having a smaller
enhancement compared to non-absorbing coatings (Lack and Cappa, 2010). However, the extent of these impacts remains uncertain, and further research is necessary to better constrain the representation of BC in models. This requires measurements of the mixing state, coating material composition, and optical properties of soot particles in the ambient atmosphere.

Aqueous phase chemical reactions occurring in aerosol liquid water (ALW) and cloud/fog droplets can contribute to the formation of particulate matter and ultimately lead to degraded air quality. One example is the partitioning of water-soluble
organic gases into the aqueous phase, where they can react to produce highly oxidized, low volatility compounds. These compounds, generally referred to as aqueous-phase SOA (aqSOA), can remain in the particle phase upon water evaporation (Bianco et al., 2020; Ervens et al., 2011, 2018; Gilardoni et al., 2016; Kim et al., 2019; Sun et al., 2010; Tomaz et al., 2018). The concentrated solute conditions of ALW promote the formation of oligomers and other high molecular weight compounds, while the higher water content conditions found in fog/cloud droplets favor the production of smaller carboxylic acids such as
acetic, formic and oxalic acid (Charbouillot et al., 2012; Ervens et al., 2011; Lim et al., 2010). After water evaporation, the residual material is internally mixed with any included BC, or can coagulate with BC particles, resulting in soot particles with high non-BC content and larger particle sizes (Collier et al., 2018; Meng and Seinfeld, 1994). The occurrence of aqueous-phase reactions are therefore expected to have a significant impact on the thickness and composition of BC coatings, ultimately impacting the optical properties of the soot aerosol (Cao et al., 2022; Cappa et al., 2019). However, a thorough understanding
of how aqueous-phase reactions affect ambient soot particles and their optical properties is still lacking.



The San Joaquin Valley (SJV) of California provides an ideal location for studying the impacts of aqueous-phase reactions and fog processing on anthropogenic soot particles (Chen et al., 2018; Collett et al., 1998; Collier et al., 2018; Ge et al., 2012a; Herckes et al., 2015; Kim et al., 2019). This region experiences high levels of humidity and frequent radiation fog events during winter (Collett et al., 1998; Herckes et al., 2015). Moreover, the SJV faces significant air pollution challenges, with severe wintertime particulate matter (PM) pollution episodes and high concentrations of black carbon due to a combination of elevated anthropogenic emissions and stagnant meteorology (Chen et al., 2018, 2020; Ge et al., 2012b; Parworth et al., 2017; Prabhakar et al., 2017; Sun et al., 2022; Watson et al., 2021; Young et al., 2016). Previous studies using aerosol mass spectrometry (AMS) have demonstrated that particulate matter less than 1µm in diameter ($PM_1$) during winter in the SJV is strongly influenced by combustion sources, including residential woodburning (RWB) and vehicle exhaust (Betha et al., 2018; Chen et al., 2018; Ge et al., 2012b; Sun et al., 2022; Young et al., 2016). Furthermore, the abundant volatile organic compounds (VOCs) co-emitted from these sources contribute to the formation of secondary organic aerosols (SOA) through both gas-phase and condensed-phase reactions (Chen et al., 2018; Ge et al., 2012b; Kim et al., 2019; Lurmann et al., 2006; Young et al., 2016).

In this study, we investigated the aqueous-phase processing of soot particles using a Soot-Particle Aerosol Mass Spectrometer (SP-AMS) deployed in Fresno, which is the largest city in the SJV with a population of approximately 500,000. In particular, we compared a multi-day fog event (high-fog) to a period with less fog (low-fog) to examine how fog droplet processing influences the chemical, physical and optical properties of soot aerosol. The SP-AMS was modified by removing the thermal vaporizer, leaving it solely equipped with the laser vaporizer. This setup allowed us to selectively measure BC-containing particles and analyze the refractory BC (rBC) and chemical composition of associated inorganic and organic coatings in real-time. Positive matrix factorization analysis of SP-AMS spectra was used to evaluate the importance of aqueous-phase reactions within fog droplets in shaping the properties of soot particles. SP-AMS ion fragments unique to fog processing were identified, offering valuable information for future ambient measurements aimed at identifying aerosol that has undergone fog/cloud processing. These results provide new insights into the role of aqueous processing in influencing the characteristics of BC-containing particles.

## 2. Experimental Methods

### 2.1 Sampling site and Instrumentation

Measurements were collected between December 19, 2014 and January 13, 2015 at the University of California Cooperative Extension (36°48'35.26″N, 119°46'42.00″W) in Fresno, CA. An Aerodyne SP-AMS was used to obtain the size-resolved concentration of refractory black carbon (rBC) and associated coating material at 5-minute time resolution. The instrument is similar in design to the Aerodyne high-resolution time-of-flight aerosol mass spectrometer (HR-AMS) but includes a 1064 nm Nd:YAG intracavity laser vaporizer for the vaporization of absorbing material such as rBC (DeCarlo et al., 2006; Onasch et al., 2012). The tungsten thermal vaporizer was physically removed from the instrument in order to only



measure aerosol species (i.e., organics, nitrate, sulfate, chloride, ammonium) mixed with rBC. Further details of the sampling setup, operation of the SP-AMS and the configuration of collocated instruments are described in section S1.1 and reported in

Collier et al. (2018) and Cappa et al. (2018).

Aerosol absorption was measured at 405 and 532 nm using a dual wavelength cavity ringdown/photoacoustic spectrometer (CRD-PAS) and 870nm using a photoacoustic extinctiometer (PAX). Details on the processing of the optical measurements can be found in Cappa et al. (2018) and in section S1.2.

### 2.2 SP-AMS Data Analysis and Source Apportionment via ME-2

SP-AMS measurements were processed in the Squirrel (V. 1.57) and PIKA (V. 1.16) analysis toolkits within the Igor Pro environment. The concentration of rBC was calculated using the sum of $C_1^+$-$C_{10}^+$. Organic species also contribute to the signal at $C_1^+$, therefore rBC contribution to $C_1^+$ was constrained using the ratio of $C_1^+/C_3^+$ measured for regal black during instrument calibration. High resolution peak fitting was performed in a similar manner to Collier et al. (2018) with the following modifications. Additional nitrogen-containing organic fragments were included in the ion list. The following

refractory metals were also fit: $K^+$, $Rb^+$, $K_2Cl^+$, $K_2NO_3^+$, and $K_3SO_4^+$. Each of these compounds have a significant negative mass defect with few other possible molecular formulae at the exact m/z ratio, allowing for their unambiguous identification. Organic aerosol elemental ratios including the molar ratios of oxygen-to-carbon (O/C) and hydrogen-to-carbon (H/C) reported here are calculated using the improved ambient method (Canagaratna et al., 2015b).

Positive Matrix Factorization (PMF) analysis was performed on the SP-AMS data matrix using the multilinear engine

2 (ME-2) solver within the Source Finder (SoFi) software (Canonaco et al., 2013; Paatero, 1999; Paatero and Tapper, 1994). The PMF factors associated with rBC aerosol resolved in this study are notated with a "BC" subscript to differentiate them from the NR-PM$_1$ factors identified in Chen et al (2018). Inputs for PMF included the high-resolution (HR) organic fragments between *m/z* 12-150, unit mass resolution (UMR) signal between *m/z* 151-307 and major inorganic ion fragments. Specifically, $C_2^+$ through $C_5^+$ were included for rBC, $NO^+$ and $NO_2^+$ for nitrate, $Cl^+$ and $HCl^+$ for chloride, $SO^+$, $SO_2^+$, $HSO_2^+$, $SO_3^+$, $HSO_3^+$

and $H_2SO_4^+$ for sulfate, $NH^+$, $NH_2^+$ and $NH_3^+$ for ammonium, and $K^+$ and $Rb^+$ for metals. Inclusion of inorganic fragments in PMF analysis constrains the rotational ambiguity of the solution and provides additional insight into the physical meaning of the factors (Sun et al., 2012; Zhou et al., 2017). For more information about ME-2 analysis please refer to section S1.3.

To calculate the species dependent mass within each BC-containing particle type, the fragments corresponding with organics, nitrate, sulfate, chloride, ammonium, rBC and K were segregated and scaled by the species dependent relative

ionization efficiency (RIE) and the coating thickness dependent collection efficiency (CE) as described in Collier et al. (2018). The RIE values for nitrate, sulfate and ammonium were determined using the thermal vaporizer, and the RIE using the laser vaporizer may vary slightly from this. An RIE of 2.9 was used for potassium (Drewnick et al., 2006) while Rb and potassium containing salts are presented in nitrate equivalent concentration. The potassium RIE was derived using a ToF-AMS equipped with a thermal vaporizer and it is possible the laser vaporizer RIE is varies from this.





### 3. Results and Discussion

#### 3.1 Soot Aerosol Composition and Properties in Fresno and the Influence of Fog Processing

The meteorological conditions during the sampling period were typical of winter in the San Joaquin Valley and were characterized by cool and humid weather with average (± σ) temperature of 9.9 ± 4.7°C and relative humidity (RH) of 83 ± 16% (Fig. S4a). The average wind speed was low (0.54 ± 0.32 m/s), indicating stagnant conditions that limited the dispersion of pollutants. Elevated rBC concentration was observed throughout the campaign, with an average of 1.04 ± 0.77 µg m$^{-3}$ (Fig. S4c), due to a prevalence of combustion sources in this area.

Between Jan 7$^{th}$ and Jan 13$^{th}$, 2015, a multiday fog event occurred, providing an opportunity for us to study the effect of fog processing on aerosol composition and properties. The visibility measured at the Fresno Yosemite International Airport, located approximately 10 km away from the sampling site, clearly indicates the persistent nature of the fog event, as the visibility remained below four km for seven consecutive days (Fig. 1, Fig. S4b). In contrast, the preceding, low-fog period between Dec 20$^{th}$ and Jan 4$^{th}$, 2015 experienced colder and drier conditions. Fog events were less frequent during this period, and when they did occur (e.g., on Dec 21$^{st}$, Dec 23$^{rd}$ and Dec 31$^{st}$ 2014), they were relatively short-lived, with visibility values below four km for 8 to 14 hours for each occurrence. However, despite the reduced occurrence of fog events, the RH reached 100% nearly every night, suggesting favorable nighttime conditions for the occurrence of aqueous-phase processing within aerosol liquid water (ALW).

A key feature of our study was that the SP-AMS was configured to exclusively measure rBC-containing aerosols. This allowed us to utilize the ratio of the total mass of inorganic and organic material to the mass of rBC (R$_{Coat/BC}$) as a metric for estimating the thickness of coatings on soot particles. In addition, because only 17% of total PM$_1$ mass was associated with rBC (Chen et al., 2018; Collier et al., 2018), the background concentrations of secondary species in our measurements were lower in comparison to the aggregated aerosols allowing for enhanced sensitivity in detecting subtle compositional changes arising from the production of secondary aerosol through aqueous-phase processing.

To gain insights into the changes associated with secondary aerosol formation during fog processing and understand the intricate chemical transformations occurring within the soot aerosol population, we compare the size-resolved chemical composition of submicrometer particles containing BC (PM$_{1,BC}$) between the high-fog period (Jan 7- Jan 13, 2015) and the low-fog period (Dec. 20 – Jan 4). A PM$_{2.5}$ cyclone was included in the sampling line prior to a diffusion dryer preventing the direct sampling of liquid fog droplets larger than 2.5 µm. Instead, the aerosol composition measured during fog events represents the composition of interstitial particles. As illustrated in Fig. 1c, the PM$_{1,BC}$ concentration increased substantially from 2.6 ± 2.0 µg m$^{-3}$ during the low-fog period to 5.1 ± 1.3 µg m$^{-3}$ during the high-fog period, accompanied by significant changes in the soot aerosol composition. A t-test revealed that the concentrations of directly emitted species, including rBC (p < 0.01) and gas-phase CO (p < 0.05) displayed statistically significant increases during the high-fog period (Fig. 1). This increase is likely associated with the accumulation of pollutants from combustion sources facilitated by the stagnant, windless conditions and decreased boundary layer height. The low-fog period also coincided with the winter holidays, and it is possible



that the emission patterns of primary sources, such as residential wood-burning and vehicle traffic varied between the two periods.

The $R_{Coat/BC}$ value increased by a statistically significant amount from $2.30 \pm 0.86$ during the low-fog period to $2.92 \pm 0.45$ during the high-fog period ($p < 0.01$; Fig. 1c). The increase of $R_{Coat/BC}$ during the fog event was attributed to the accumulation of secondary species, including nitrate, sulfate, ammonium and oxidized organics (Fig. 1c, S4). These observations indicate an enhanced production of secondary aerosol species facilitated by aqueous-phase reactions within fog droplets and underscore the profound influence of fog processing on aerosol composition and physical properties.

Nitrate exhibited the most significant increase on soot particles during the high-fog period, with the concentration of $NO_{3, BC}$ rising by a factor of five, from $0.28 \pm 0.14$ µg m$^{-3}$ during the low-fog period to $1.29 \pm 0.31$ µg m$^{-3}$ (Fig. 1c). Its contribution to the $PM_{1,BC}$ mass also increased considerably from 11% to 25% (Fig. 1d, 1e). These observations provide clear evidence that the presence of fog droplets promoted the formation of nitrate on BC particles. Particulate nitrate in the atmosphere is formed via multiple pathways. One of the pathways involves the gas-phase reaction between $NO_2$ and the OH radical, resulting in the formation of $HNO_3$, a highly water-soluble compound which rapidly partitions into liquid droplets (Finlayson-Pitts and Pitts, 1997). Upon the evaporation of water, the abundant ammonia present in the SJV region inhibits the evaporation of nitrate.

Another pathway for nitrate formation involves the heterogeneous uptake and subsequent hydrolysis of $N_2O_5$. $N_2O_5$ is formed through the reaction of $NO_2$ and $NO_3$ radical, which itself is formed by the reaction between $NO_2$ and $O_3$

(Ravishankara, 1997). As both $N_2O_5$ and its precursor, the $NO_3$ radical, are quickly photolyzed during the daytime, nitrate formation via the $N_2O_5$ pathway is typically considered important only at night while the OH pathway dominates during the day. However, previous studies have demonstrated that during fog/cloud events, the reduced transmission of solar radiation, coupled with an elevated droplet surface area, can make $N_2O_5$ hydrolysis a significant source of nitrate even during the daytime (Brown et al., 2016; Wu et al., 2021; Zhang et al., 2022). Thus, the suppressed solar radiation during the high-fog period may

result in elevated daytime steady-state concentrations of $NO_3$ radical and $N_2O_5$. Furthermore, the suppressed solar radiation will also likely lead to decrease in the steady state concentration of the OH radical, further reducing the role of the OH pathway. Together, this suggests that the $N_2O_5$ formation pathway could be an important factor contributing to the elevated concentrations of nitrate seen during the high-fog period.

The concentration of sulfate was also enhanced in soot aerosols during the high-fog period, exhibiting a 2.5-fold

increase compared to the low-fog period (Fig. 1). Since $SO_2$ concentrations were comparable between the two periods, aqueous-phase reactions occurring within fog droplets likely played a role in the increasing sulfate concentration during the high-fog period. This observation aligns with the well-established understanding that $SO_2$ undergoes rapid conversion to $SO_4^{2-}$ through aqueous-phase reactions (Seinfeld and Pandis, 2006).

Organic compounds were the most abundant species on soot aerosol, contributing 38% and 48% of $PM_{1,BC}$ mass

during the high-fog period and low-fog period, respectively. The $Org_{BC}$ was significantly higher during the high-fog period, increasing from $1.25 \pm 1.13$ µg m$^{-3}$ to $1.96 \pm 0.57$ µg m$^{-3}$. Additionally, the composition of $Org_{BC}$ differed between the two



periods showing evidence of fog-induced increases of oxygenated organic species in soot aerosols (Fig. 1c). The detailed discussion of this process is provided in the subsequent sections.

Another important species present in $PM_{1,BC}$ in Fresno is potassium, with an average concentration of $53 \pm 50$ ng m$^{-3}$ and accounting for between 0.7% and 4% of $PM_{1,BC}$ (Fig 2). $K^+$ is emitted directly from the combustion of biomass and serves as an inert tracer for biomass burning emissions as, unlike levoglucosan, $K^+$ does not degrade during atmospheric processing (Andreae, 1983). However, there is significant variation of the chemical form of potassium in the atmosphere. Although $K^+$ is usually emitted as KCl or KOH, these compounds undergo rapid acid displacement reactions with $H_2SO_4$ and $HNO_3$ to form $K_2SO_4$ and $KNO_3$ (Cao et al., 2019; Li et al., 2013; van Lith et al., 2008; Sorvajärvi et al., 2014). KCl, $K_2SO_4$ and $KNO_3$ have been identified previously in aerosols within BB plumes (Li et al., 2003, 2010) and have been detected with mass spectrometry as $K_2Cl^+$ (*m/z* 112.896), $K_2NO_3^+$ (*m/z* 139.915), and $K_3SO_4^+$ (*m/z* 212.843) (Pratt et al., 2010; Shen et al., 2019; Wang et al., 2020). Similarly, $Rb^+$ is also associated with BB emissions (Artaxo et al., 1994; Rivellini et al., 2020). Quantification of $K^+$ is generally challenging within the AMS as potassium can undergo thermal ionization on the vaporizer surface, leading to highly uncertain ionization efficiency (Drewnick et al., 2015). Even in the absence of the thermal vaporizer, it is possible that potassium can undergo thermal ionization upon the BC particle surface during the vaporization process. However, previous studies have demonstrated a strong correlation between the K signal measured by an AMS and the $K^+$ concentration measured by collocated ion chromatography attached to a Particle Into Liquid Sampler (PILS), indicating that the AMS-measured K signals can be effectively utilized to determine the temporal variations in particulate potassium concentration (Parworth et al., 2017).

The dominant form of potassium measured was $K^+$, which exhibited a strong correlation with BBOA ($r^2 = 0.93$) and $Rb^+$ ($r^2 = 0.78$). However, $K_2Cl^+$, $K_2NO_3^+$ and $K_3SO_4^+$ were also unambiguously identified (Fig. 2). The concentration of $K_2Cl^+$ is sporadic and increased substantially during periods influenced by fresh RWB emissions. Interestingly, $K_2Cl^+$ concentration remained low throughout the high-fog period, suggesting that the presence of fog droplets may facilitate the physical or chemical removal of KCl. In contrast, $K_3SO_4^+$ concentrations were elevated during the high-fog period compared to the low-fog period. The correlations between $K_3SO_4^+$ and $SO_{4,BC}$ concentrations showed variable slopes, gradually decreasing over the course of the high-fog period (Fig. 2b). The concentration of $K_2NO_3^+$ was considerably lower than the other forms of potassium, despite the abundance of nitrate in the SJV. Parworth et al (2017) found low concentrations of gas-phase $HNO_3$ likely resulting from the rapid formation of ammonium nitrate due to the high ammonia concentrations. Hence, the low concentrations of $K_2NO_3$ may be attributed to insufficient $HNO_3$ levels to form significant amounts of $KNO_3$ through acid replacement reactions.

### 3.2 Sources and Organic Coating Processing of Soot Particles in Fresno

Through source apportionment analysis that included both organic and inorganic species, we identified four soot particle types. These included a biomass burning organic aerosol ($BBOA_{BC}$) associated with residential wood burning activity, hydrocarbon like organic aerosol ($HOA_{BC}$) related to vehicle emissions, fog-related oxidized organic aerosol ($FOOA_{BC}$)



connected with aqueous phase processing occurring with fog droplets, and winter-background oxidized organic aerosol (WOOA$_{BC}$) associated with both aqueous phase and photochemical processes. The first of these two (BBOA$_{BC}$ and HOA$_{BC}$) are named according to their primary emission source, while the FOOA$_{BC}$ and WOOA$_{BC}$ are named based on the pathways by which the BC was coated and processed following emission. Biomass burning and vehicles are believed to be the original sources of the BC in FOOA$_{BC}$ and WOOA$_{BC}$, however the primary source signature has degraded or been overshadowed by

secondary aerosol formation during atmospheric processing.

Among these, BBOA$_{BC}$ exhibited the highest concentration, with an average concentration of 0.55 µg m$^{-3}$, accounting for 37% of the OA$_{BC}$ mass over the duration of the campaign. As shown in Fig. 3a, the BBOA$_{BC}$ spectrum had significant contributions from the C$_2$H$_4$O$_2^+$ (*m/z* = 60) and C$_3$H$_5$O$_2^+$ (*m/z* = 73) fragments, both of which are markers for anhydrous sugars such as levoglucosan released during the combustion of biomass (Aiken et al., 2010; Alfarra et al., 2007; Cubison et al., 2011).

The organic mass in this factor showed a fraction of signal at *m/z* 60 ($f_{60}$) of 3.0%, and an O/C of 0.46 indicating BBOA$_{BC}$ likely represents freshly emitted soot particles from BB. Furthermore, Fig. 3a shows that BBOA$_{BC}$ had little contribution from inorganic compounds except for K$^+$, which accounted for 3.4% of the total factor mass. Additionally, BBOA$_{BC}$ displayed the lowest R$_{Coat/BC}$ (1.49) among the four factors, suggesting that fresh wood smoke soot aerosol at this site was predominantly thinly coated.

The BBOA$_{BC}$ concentration was higher during the low-fog period, which can be attributed to an increased residential wood burning (RWB) activity due to the colder temperatures and the winter holidays (Fig. 1c, Fig. 2a). It is also possible that the lower BBOA$_{BC}$ concentration during the high-fog period was due to rapid fog processing of fresh wood smoke aerosol into OOA, removing the BB source signatures. The time series analysis shows comparable diurnal patterns of BBOA$_{BC}$ concentrations during both periods, with peak concentrations occurring between 19:00-24:00 and a minor increase observed

between 7:00-10:00 (Fig. S6a). These results are consistent with previous observations of prevalent RWB activity in Fresno during the winter season (Betha et al., 2018; Ge et al., 2012b; Sun et al., 2022; Young et al., 2016).

The HOA$_{BC}$ factor likely represents soot particles emitted by vehicle traffic and accounted for 18% of OA mass on BC particles. Its mass spectrum was primarily composed of C$_x$H$_y^+$ ions, displaying an enhancement of the C$_n$H$_{2n-1}^+$ and C$_n$H$_{2n+1}^+$ ion series (Fig. 3) which is characteristic of fossil fuel combustion (Canagaratna et al., 2004; Collier et al., 2015;

Zhang et al., 2005). HOA$_{BC}$ was the least oxidized factor identified in this study, with an O/C of 0.30. It displayed a slightly elevated contribution from inorganic nitrate, which accounted for 3% of the total factor mass, possibly due to the rapid oxidation of NO$_x$ co-emitted from vehicles. Black carbon accounts for 34% of the total factor mass, yielding an average R$_{Coat/BC}$ of 1.95.

The average organic concentrations of HOA$_{BC}$ during the low-fog and high-fog periods were 0.30±0.32 µg m$^{-3}$ and

0.24±0.23 µg m$^{-3}$, respectively, indicating relatively consistent emissions of soot particles from vehicular sources throughout the campaign. However, the diurnal profiles of HOA$_{BC}$ showed notable differences between the two periods (Fig. S6b). During the high-fog period, two diurnal peaks of similar magnitudes were observed. The peak in the morning (09:00-10:00) corresponds to rush hour, while the peak in the evening (19:00-23:00) is influenced by a combination of rush hour, decreasing



boundary layer height and other, late night combustion activity. In contrast, during the low-fog period, the morning rush-hour
peak was nearly absent and the evening peak occurred two hours earlier. These differences may be due to different traffic
patterns during the winter holidays, or differences in boundary layer height between the two periods.

The two $OOA_{BC}$ factors showed distinct spectral and temporal features, suggesting that they may represent soot
particles processed through different atmospheric pathways, with coatings formed via different precursor sources or secondary
formation processes. The concentration of $FOOA_{BC}$ showed a dramatic increase during the high-fog period, rising by over a
factor of 10 from $0.18 \pm 0.13$ µg m$^{-3}$ to $2.2 \pm 0.5$ µg m$^{-3}$ (Fig. 1). In contrast to this, $WOOA_{BC}$ did not exhibit significantly
different concentrations between the two periods. Overall, the organic matter in the $FOOA_{BC}$ factor was less oxidized than the
$WOOA_{BC}$ factor, with O/C ratios of 0.50 and 1.01, respectively. The O/C ratio of the $FOOA_{BC}$ is comparable to the O/C ratios
observed in residual particles formed from the atomization of fog waters collected in Fresno (Kim et al., 2019). The $FOOA_{BC}$
spectrum is dominated by the $C_2H_3O^+$ fragment and shows a stronger influence from $C_xH_y^+$ fragments (Fig. 3). On the other
hand, the $WOOA_{BC}$ factor has a higher contribution of oxygen-containing fragments such as the $CO_2^+$ at *m/z* 44. Furthermore,
the $WOOA_{BC}$ factor also exhibits an enhanced fraction of organic mass at *m/z* 29, dominated by the $CHO^+$ fragment (Fig. 3),
a marker that has been previously associated with aqSOA (Gilardoni et al., 2016). Nearly all of the nitrate, sulfate and
ammonium mass is attributed to the $OOA_{BC}$ factors, indicating the significant contribution of secondary inorganic species as
coating material on processed BC aerosol in Fresno. Nitrate and sulfate accounted for 32% and 2% of the total $FOOA_{BC}$ mass
and 24% and 3% of the total $WOOA_{BC}$ mass.

The diurnal profile of $FOOA_{BC}$ shows relatively little variation during the high-fog period, with a slight decrease
observed at 08:00, and a maximum at 13:00 (Fig. S6c). This decrease in the morning coincided with the lowest visibility
values and may be attributed to fog scavenging, or the growth of particles beyond the 1 µm transmission range of the SP-AMS.
Based on our findings, we deduce that the organic matter in $FOOA_{BC}$ represents SOA formed through aqueous phase
processing within liquid fog droplets.

The $WOOA_{BC}$ concentration shows notably different diurnal profiles between the two periods (Fig S6d). Although
the concentration is flat during the low-fog period, this factor showed a notable daytime peak between 08:00 and 20:00 during
the high-fog period, indicating the role of photochemical reactions in the formation of $WOOA_{BC}$. During the low-fog period,
aqueous-phase processes likely occurred primarily in concentrated aerosol liquid water (ALW) under subsaturated conditions
rather than in relatively dilute droplets under foggy conditions. We hypothesize that $WOOA_{BC}$ represents a general OOA factor
internally mixed with BC, influenced by SOA formed through aqueous processing in ALW, as well as gas phase photo-
oxidation.

Although solar radiation was slightly reduced during the high-fog period compared to the low-fog period, the mixing
ratios of gas-phase $O_3$ and odd oxygen ($O_x = O_3 + NO_2$ (Herndon et al., 2008; Wood et al., 2010)) were similar between the
two periods, suggesting comparable photochemical activity (Fig. S11). The role of photochemistry in SOA production was
further investigated using the correlation of the aerosol species and $O_x$ (Fig. S9). Overall, there were low correlations ($r^2 <$
0.2) between the concentration of both $OOA_{BC}$ factors and $O_x$, suggesting that photochemical processes were unable to explain



all SOA formation. When normalized to the rBC concentration, a slight increase in $WOOA_{BC}/rBC$ was seen at high $O_x$ values during the daytime ($r^2 = 0.32$), supporting that $WOOA_{BC}$ production was influenced by daytime photochemical processes

during the low-fog period. However, the concentrations of nitrate and sulfate showed negligible changes with varying $O_x$ levels, supporting the important contribution of aqueous phase reactions in the production of these species.

The soot particles represented by the $FOOA_{BC}$ and $WOOA_{BC}$ factors displayed larger $R_{Coat/BC}$ than the soot particles represented by the $BBOA_{BC}$ and $HOA_{BC}$ factors, with $R_{Coat/BC}$ values of 3.50 and 3.27, respectively. These larger values are consistent with the accumulation of secondary material onto BC particles or the coagulation with non-BC containing secondary

particles during atmospheric processing. However, the $R_{Coat/BC}$ observed in our study is considerably lower than values reported in studies that analyzed heavily processed $PM_1$ using laser-only SP-AMS, where $R_{Coat/BC}$ values were found to exceed 10 (Healy et al., 2015; Lee et al., 2017; Wang et al., 2017). Therefore, we infer that $FOOA_{BC}$ and $WOOA_{BC}$ likely represent BC particles coated with secondary materials that have undergone limited processing.

Both $OOA_{BC}$ factors also show a minor contribution from K and chloride in the coating material (Fig. 3, Fig. S5),

indicating that RWB is a notable source of the processed soot particles at this site. This hypothesis is supported by the slightly elevated $f_{C2H4O2}$ signal observed in the mass spectra of both $FOOA_{BC}$ and $WOOA_{BC}$ (0.74% and 0.84%, respectively). To estimate the influence of processed BB emissions on the two $OOA_{BC}$ factors, we utilized K as an inert tracer. The K mass concentration associated with the $FOOA_{BC}$ and $WOOA_{BC}$ factors was multiplied by the BC/K ratio observed in the $BBOA_{BC}$ factor. Assuming a constant BC/K ratio for BB aerosol during atmospheric processing, this calculation resulted in a BB-

influenced (BB-inf) fraction of 43% of the BC in $FOOA_{BC}$ and 49% of the BC in $WOOA_{BC}$. The ratio of BB-inf $OOA_{BC}$ to BBOA was significantly higher (p < 0.01) during the high-fog period than the low-fog period, suggesting that the presence of droplets coupled with the stagnant conditions resulted in a higher abundance of processed BBOA (Fig S10). These results also highlight the importance of RWB emissions to the total BC budget in this location.

Due to the abundant $NH_{3(g)}$ emissions in SJV, aerosols are typically neutralized (Parworth et al., 2017; Young et al.,

2016). Interestingly, the $BBOA_{BC}$, $HOA_{BC}$ and $FOOA_{BC}$ factors all exhibited a ratio of measured ammonium to predicted ammonium ($NH_{4,meas}/NH_{4,pred}$) less than 1, with ratios of 0.80, 0.93 and 0.96 respectively, indicating an apparent deficit of ammonium. Although this may suggest the presence of acidic aerosol (Zhang et al., 2007), it is more likely due to the presence of the potassium salts of sulfate, nitrate and chloride. In contrast to this, the calculated $NH_{4,meas}/NH_{4,pred}$ ratio for the $WOOA_{BC}$ factor is 1.47, indicating an excess of ammonium associated with this factor. This excess may indicate the presence of organic

acids, as suggested by the elevated $CO_2^+$ and $CHO_2^+$ signals, which have been proposed as tracers for organic acids (Jiang et al., 2021; Sorooshian et al., 2010; Yatavelli et al., 2015). Additionally, as the organic matter in $WOOA_{BC}$ has the highest N/C ratio (0.032) among all factors resolved in this study, the elevated ammonium signal could also be contributed by amino compounds, which can fragment into $NH_x^+$ ions within the AMS (Ge et al., 2014).

Chen et al. (2018) conducted source apportionment analysis of NR-$PM_1$ using a co-located HR-ToF-AMS, and a

comparison with their results provides insight into the differences between the bulk NR-$PM_1$ composition and the black carbon containing fraction. Four PMF factors were also identified for the source apportionment of NR-$PM_1$ including $BBOA_{NR-PM1}$



(O/C = 0.47, H/C = 1.7), $HOA_{NR-PM1}$ (O/C = 0.11, H/C = 1.8), Nitrate-related OOA ($NOOA_{NR-PM1}$; O/C = 0.44, H/C = 1.7) and very oxygenated OA ($VOOA_{NR-PM1}$; O/C = 0.78, H/C = 1.7) (Chen et al., 2018). Due to the two vaporization methods resulting in different fragmentation patterns (Avery et al., 2020; Canagaratna et al., 2015a) the temporal features of each factor are

compared rather than a direct comparison of the spectra. The scatter plots of the time series (not shown) reveal a strong correlation between $BBOA_{BC}$ and $BBOA_{NR-PM1}$, with an $r^2$ value of 0.81 and a slope of 0.14 obtained through orthogonal distance regression with an intercept of zero. Likewise, the correlation between $HOA_{NR-PM1}$ and $HOA_{BC}$ was high ($r^2 = 0.70$). Therefore, the two primary OA factors, HOA and BBOA were combined and displayed in Figure 4b. There was no statistical difference in the slope of $\Sigma POA_{BC}$ and $\Sigma POA_{NR-PM1}$ during the two periods. Overall, approximately 12% of POA mass was

mixed with rBC in $PM_1$ during this study assuming the same RIE in the laser and tungsten vaporizer.

The temporal behavior of the OOA factors differ dramatically between the $NR-PM_1$ and $PM_{1,BC}$ solutions (Fig. 4a), suggesting that atmospheric processing may have impacted the non-BC containing fraction differently than the BC containing fraction. The time series of $FOOA_{BC}$ shows a strong correlation with $NOOA_{NR-PM1}$ ($r^2 = 0.80$) and no correlation with $VOOA_{NR-PM1}$ ($r^2 = 0.01$). However, the $NOOA_{NR-PM1}$ concentration also shows periods of elevated concentrations during the low-fog

period signifying that $NOOA_{NR-PM1}$ is not unique to fog processing. Cappa et al. (2018) argues that this factor may be formed through various nocturnal oxidation pathways including aqueous-phase reactions and nitrate radical chemistry. The $WOOA_{BC}$ time series shows no correlation ($r^2 < 0.1$) with either $OOA_{NR-PM1}$ factor. The correlation between $\Sigma OOA_{BC}$ and $\Sigma OOA_{NR-PM1}$ is moderate during both periods, however the slope increases from 0.055 in the low-fog period to 0.12 in the high-fog period and the difference is statistically significant ($p < 0.01$). The steeper slope in the presence of fog suggests that droplet processing

creates a more homogenous aerosol population, with a larger portion of secondary material internally mixed with rBC.

Approximately 4% of total sulfate is internally mixed with rBC (Fig. 4c). The percent of sulfate mixed with black carbon is slightly enhanced during the low-fog period (4.6% of total sulfate in $PM_1$ vs. 3.9% during high-fog period; $p < 0.05$). Although the concentration of $NO_{3,BC}$ increased by a factor of five during the high-fog period as discussed above, the fraction mixed with rBC is still low during the high-fog period (8.6% of total $NO_3$ in $PM_1$, in contrast to 6.4% during the low-fog

period; $p < 0.01$).

### 3.3 Effect of Fog Processing on Soot Particle Size Distribution

The size distribution of soot particles changed significantly over the course of the campaign, peaking at larger sizes (600-700 nm in $D_{va}$) during the high-fog event consistent with cloud/fog processing (Fig. 5; Collier et al., 2018; Eck et al., 2012; Ge et al., 2012a). The average size distribution of each PMF factor was calculated by performing linear decompositions

of the average size resolved mass spectra (Fig. 5), following the method described by Ge et al., (2012b). To examine the effects of fog processing, the decomposition was performed separately for the low-fog and high-fog periods. The $HOA_{BC}$ and $BBOA_{BC}$ factors show slightly different size distributions between the two periods, with the size mode for each factor shifting to smaller diameters during the high-fog period (Fig. 5a-b). Specifically, the BBOA size distribution shifts from a peak at 300nm to 180nm in $D_{va}$, while the HOA size distribution shifts from a broad peak ranging from 200nm to 400nm to a narrower



peak between 100nm and 300nm $D_{va}$.  Additionally, during the low-fog period both $HOA_{BC}$ and $BBOA_{BC}$ size distributions extended to particle sizes larger than 500nm $D_{va}$, while this tail is completely absent during the high-fog period.  As the size distributions of freshly emitted POAs are likely similar between the two periods, the observed shift towards smaller sizes during the high-fog period may be attributed to rapid wet deposition or droplet scavenging, which selectively removes larger particles from the aerosol population.

The $FOOA_{BC}$ size distribution shows a single mode, peaking at 700nm, which is consistent with droplet processing. On the other hand, the $WOOA_{BC}$ factor appears to contribute more to smaller accumulation mode particles compared to the $FOOA_{BC}$ factor.  This behavior could be due to the condensation of SOA species produced in the gas phase through photochemical processes.

Due to the changes in the aerosol size distributions discussed above, the size dependent $PM_{1,BC}$ composition varied 375    significantly between the two periods (Fig. 5e, f).  During the high-fog period, small soot particles with $D_{va} < 200$nm are primarily composed of rBC and the POA species.  Larger particles are predominantly coated with $FOOA_{BC}$ and ammonium nitrate.  Interestingly, the coating thickness exhibits a strong size dependence, increasing from $R_{coating/BC} < 1$ at 100nm to nearly 6 at 1μm.  In contrast, the composition was less variable across the size range during the low-fog period.  The rBC fraction remained relatively constant for different sized particles, yielding only a minor increase in coating thicknesses at larger particle 380    sizes.

**3.4. aqSOA Formation on Soot Particles**

The prolonged fog event and high concentrations of liquid water during the high-fog period provide an opportunity to explore the formation of aqSOA in the ambient atmosphere.  Table 1 lists the changes in the concentration of HR-AMS ion fragments related to different sources, including organic acids, methanesulfonic acid, BBOA and vehicle emissions.

Methanesulfonic acid (MSA, $CH_3SO_3H$) or its salt – mesylate ($CH_3SO_3^-$) is often used as a marker for aqueous phase processing and has been previously identified in aerosol and fog water in Fresno (Ge et al., 2012b; Kim et al., 2019).  Here, two previously identified AMS marker ions for MSA, $CH_2SO_2^+$ and $CH_3SO_2^+$, are observed to be well separated from adjacent ions and are used to estimate the $MSA_{BC}$ concentration.  The slope of the signal intensities of these two ions was 0.37 ($r^2 = 0.71$), which is similar to the ratio seen for pure MSA sampled in the laboratory and suggests that these fragments are primarily 390    produced by MSA (Ge et al., 2012a).  The total concentration of MSA in soot particles was estimated using the following formula:

$$MSA_{BC} = \left(CH_2SO_{2,BC}^+ + CH_3SO_{2,BC}^+\right) / 0.119 \tag{1}$$

Where $CH_2SO_2^+{}_{BC}$ and $CH_3SO_2^+{}_{BC}$ are the measured mass concentrations of these two ions and 0.119 is their fractional contribution to the total mass spectra of pure MSA reported in Ge et al, (2012a).  The $MSA_{BC}$ concentration during the low-395    fog period was $2.3 \pm 1.9$ ng m$^{-3}$ and during the high-fog period it was approximately 5 times higher ($11.6 \pm 3.3$ ng m$^{-3}$; Fig. 1c, Table 1).  Ge et al. (2012a) reported an average MSA concentration of 18 ng m$^{-3}$ in NR-PM$_1$ for the same site during winter 2010, and the lower concentration measured in our study is consistent with only capturing the MSA present in the rBC fraction.



The MSA$_{BC}$ concentration during the high-fog period corresponds to 0.59% of OA$_{BC}$ mass, similar to the 0.5% reported by Ge et al. (2012a). Only the CH$_3$SO$_2^+$ fragment was included in the PMF analysis, and this fragment was mainly attributed to the

FOOA$_{BC}$ factor (67%) and the WOOA$_{BC}$ factor (23%), with a minor contribution from BBOA$_{BC}$ (9%). This highlights the influence of aqSOA on both OOA$_{BC}$ factors.

Organic acids are an important component of OA and have been found to contribute up to 50% of OA mass in some ambient samples (Sorooshian et al., 2010; Yatavelli et al., 2015). These compounds, often highly oxidized, have the potential to enhance the CCN ability of the aerosol. In HR-AMS datasets, CO$_2^+$ (*m/z* 44) is often used as a marker for organic acids and

CHO$_2^+$ (*m/z* 45) has also been proposed as a marker fragment for these compounds (Duplissy et al., 2011; Jiang et al., 2021; Ng et al., 2011). At the study location, the CO$_2^+$ fragment shows contributions from all the PMF factors, emphasizing the diverse sources of this fragment. Indeed, 38% of CO$_2^+$ is associated with the two primary factors, indicating the presence of either directly emitted organic acids or other oxygenated functional groups that produce this fragment. In contrast, CHO$_2^+$ appears to be strongly associated with secondary processes. Nearly all of the signal from this fragment is attributed to the

OOA$_{BC}$ factors, with 35% and 62% of its measured signal apportioned to WOOA$_{BC}$ and FOOA$_{BC}$, respectively.

Here we also propose the use of the CH$_2$O$_2^+$ ion (*m/z* 46) as a novel tracer fragment for fog/cloud droplet processing. Interestingly, 98% of mass of this ion is apportioned to the FOOA$_{BC}$ factor, and a strong correlation (r$^2$ = 0.88) is seen between the timeseries of CH$_2$O$_2^+$ and FOOA$_{BC}$. Based on the SP-AMS spectra of pure oxalic acid measured in the laboratory, as well as the NIST data base spectra, we hypothesize that this fragment is generated from oxalic acid (HO$_2$C-CO$_2$H) or its conjugate

base, oxalate (C$_2$O$_4^{2-}$; Fig. S13). Oxalate is one of the most abundant atmospheric carboxylates and is mainly produced through cloud processing of water soluble compounds such as glyoxal and methylglyoxal (Collett et al., 2008; Ho et al., 2007; Miyazaki et al., 2009; Nah et al., 2018; Tan et al., 2010; Wang et al., 2010) and phenols (Jiang et al., 2021; Sun et al., 2010; Yu et al., 2014). The volatility of ammonium oxalate has previously been found to be several orders of magnitude lower than oxalic acid, making this the most likely form in the ammonia rich environment of the SJV (Paciga et al., 2014). The CH$_2$O$_2^+$ fragment

has also been identified in aqueous phase OOA PMF factors in Xi'an and Beijing (Duan et al., 2021; Sun et al., 2016) and in cloud droplets following illumination (Schurman et al., 2018). Other possible formation pathways of oxalate are the fragmentation of larger dicarboxylic acids (Ervens et al., 2011) or primary emission from biomass burning (Wang et al., 2007). In this study, we see no significant relationship between CH$_2$O$_2^+$ and BBOA$_{BC}$ (r$^2$ = 0.02). This finding is in contrast to measurements in China, where primary BB emissions accounted for 30% of the oxalate mass (Yang et al., 2014).

It is important to note that in ambient measurements, the HR-AMS signal at m/z 46 is typically dominated by NO$_2^+$ (45.9929 amu), and the ability to resolve CH$_2$O$_2^+$ (46.0055 amu) depends on the resolution of the instrument. Figure S13c displays the high-resolution peak fitting of the signal at *m/z* 46, confirming that the signal attributed to CH$_2$O$_2^+$ is robust and not an artifact resulting from the high-resolution fitting algorithm. According to the NIST database, other common mono- and di-carboxylic acids do not show significant production of this specific fragment. Formic acid is an exception; however, due

to its high volatility, this compound is not expected to be present in the particle phase.



Previous studies have used the strong correlation between oxalic acid and sulfate across various ambient conditions to assess the extent of cloud processing experienced by an air parcel (Hilario et al., 2021; Sorooshian et al., 2006; Yu et al., 2005). While sulfate production is dependent on the liquid water content of the droplets, aqSOA production, including the formation of oxalate, is limited by the uptake of oxidants to the droplet, and is therefore dependent on droplet surface area

(Ervens et al., 2014; McVay and Ervens, 2017). Figure 6 depicts the correlation between the different organic acid tracers discussed above and sulfate (Fig 6a,c,e) or nitrate (Fig 6b,d,f). Notably, $CH_2O_2^+$ shows significant enhancement and a strong correlation with both sulfate and nitrate during the high-fog period (Fig. 6a,b, Table 1). In contrast, the periods of elevated sulfate concentrations during the low-fog period are not associated with elevated concentrations of $CH_2O_2^+$. This variation in slope between the two periods suggests that while aqueous-phase processes can lead to sulfate formation during the low-fog

period, the conditions are not conducive to the formation of the parent compound of $CH_2O_2^+$. We hypothesize that this compound is instead formed through processes occurring in the droplets, which only occur during the high-fog period. Unlike sulfate, nitrate can be produced through both photochemical and heterogeneous reactions as discussed earlier. Furthermore, due the semi-volatile nature of ammonium nitrate, its concentration can be influenced by gas-particle partitioning, which is driven by atmospheric conditions and particle acidity. Despite these complications, a discernible relationship between $CH_2O_2^+$

and nitrate is still seen during the high-fog period (Fig 6b). This highlights the important role of fog processing in controlling the formation of particulate nitrate as has been seen in previous, bulk aerosol measurements in Fresno (Chen et al., 2018).

The relationship between the $CHO_2^+$ fragment and sulfate are comparable between both periods (Fig. 6a), indicating that similar processes govern the enhancement of both species. Finally, $CO_2^+$ shows a negligible overall correlation with sulfate (Fig 6e). This finding is expected as $CO_2^+$ can be produced through the fragmentation of a multitude of compounds unrelated to aqueous-phase processing. The relationship between $CHO_2^+$ and $SO_{4,BC}$ (Fig. 6c) as well as $CO_2^+$ and $SO_{4,BC}$ (Fig.

6e) during the low-fog period shows a variation in slope with time, resulting in the overall low correlation, emphasizing the different sources of these species.

**3.5. Influence of aqueous-phase processing on soot particle absorption properties and hygroscopicity**

The soot particle composition as a function of $R_{Coat/BC}$ is shown in Fig. 7. The results reveal that the BC-containing

POA factors are the dominant components when the rBC particles were thinly coated (i.e., $R_{Coat/BC} < 2$). However, as the coating thickness increased, the contributions of $FOOA_{BC}$ and nitrate become more prominent. Interestingly, the mass fraction of $FOOA_{BC}$ reaches a peak at an $R_{coat/BC}$ of 4, followed by a decrease at higher coating thickness. This suggests that during fog events, processed soot particles are likely removed from the atmosphere through wet deposition or advected away from the sampling site prior to becoming thickly coated. Instead, thickly coated soot particles ($R_{Coat/BC} > 4$) at this site are dominated

by ammonium nitrate, $HOA_{BC}$ and $WOOA_{BC}$. Previous studies have identified thickly coated HOA aerosol originating from diesel emissions, due to the rapid condensation of lubricating oils onto rBC aerosol (Carbone et al., 2019; Willis et al., 2016). Additionally, the higher mass fraction of $WOOA_{BC}$ at large $R_{coat/BC}$ values suggests the presence of particles that have undergone significant processing, possibly representing regionally transported aerosols.



Figure 7 also depicts the average rBC absorption enhancement ($E_{abs}$) for each $R_{coat/BC}$ bin. $E_{abs}$ describes the relative increase in the mass absorption coefficient (MAC) of coated BC compared to pure BC due to the lensing effect (Cappa et al., 2012; Lack and Cappa, 2010). Details on the calculation of $E_{abs}$ can be found in section S1.2 and Cappa et al. (2019). Although no clear monotonic trend of $E_{abs}$ with coating thickness is observed, there is an increase in $E_{abs}$ between $R_{coat/BC}$ of 2.5 to 4, followed by a subsequent decrease at larger $R_{Coat/BC}$ (Fig. 7). This observation suggests that the observed $E_{abs}$ is dependent not only on coating thickness, but also on the composition of the coating material. A high fraction of secondary material, particularly oxidized organic formed through fog processing, may contribute to a stronger absorption enhancement. However, it is important to consider that the particular dependence of $E_{abs}$ on $R_{coat/BC}$ could also be due to heterogeneity in the aerosol mixing state, the presence of brown carbon, or changes in the size distribution, as discussed in section 3.4. Regardless, our results agree with previous studies that have identified the critical role of secondary species in influencing $E_{abs}$ (Liu et al., 2015; Xie et al., 2019; Zhang et al., 2018; Zheng et al., 2022).

The presence of hygroscopic coating material on rBC aerosol greatly increases the ability for the soot particles to take up water. This can allow them to act as cloud condensation nuclei, and alter their impact on cloud properties. To explore this, the ALW associated with inorganic compounds present in BC-containing particles ($ALWC_{E-AIM,BC}$) was estimated using the Extended Aerosol Inorganic Model (E-AIM) (Clegg et al., 1998) and the ALW associated with organics ($ALWC_{Org,BC}$) was estimated using the parameterization introduced in Petters and Kreidenweis (2007). The total ALWC associated with BC ($ALWC_{BC}$) was calculated as the sum of these two components. A detailed description of the ALWC calculations is included in section S1.3. The average $ALWC_{BC}$ increased from $2.8 \pm 2.5$ µg m$^{-3}$ during the low-fog period to $8.5 \pm 5.6$ µg m$^{-3}$ during the high-fog period (Fig. S14). The $ALWC_{BC}$ shows a strong diurnal profile driven by the temporal variation of RH, indicating that aqueous processing within ALW may be prevalent at night throughout the entire campaign. However, the 3-fold increase that is seen during the high-fog period is a result of the high concentrations of ammonium nitrate and other hygroscopic coating material. The $ALWC_{BC}$ associated with organics (average 26%) is similar to measurements made in Beijing (30%) and other urban areas globally (21%) (Jin et al., 2020; Nguyen et al., 2016). The high concentrations of $ALWC_{BC}$ estimated here highlight that despite the hydrophobicity of the rBC core of the soot aerosol, the presence of ammonium nitrate and oxidized organic material allows soot particles to uptake water and provide sites for aqueous phase processing. Fog conditions generally have relatively low ambient supersaturation values, therefore it is necessary for particles to be sufficiently hygroscopic to initially activate and undergo droplet processing. However, following this initial activation further aqueous phase processing significantly increases the particle hygroscopicity.

## 4. Conclusions

BC-containing aerosol was sampled in a polluted, urban environment in the San Joaquin Valley of California, where residential wood burning and vehicle emissions are the main sources of soot particles. The presence of a persistent multiday fog event provided an excellent opportunity for a comprehensive case study on the effects of droplet processing on the



chemical, physical and optical properties of soot aerosol. Through source apportionment analysis, two distinct BC-containing OOA factors were resolved: one related to fog processing (FOOA$_{BC}$) and the other representing a general winter OOA (WOOA$_{BC}$) impacted by both photochemical and aqueous phase reactions. $CH_2O_2^+$ was identified as a unique HRMS fragment that was only detectable in soot particles during the high-fog period and is proposed as an HR-AMS tracer ion for fog and

cloud processing. This fragment is thought to originate from dicarboxylates, such as oxalate, which is produced through droplet processes. Although caution is advised when peak fitting is performed on this ion due to its close proximity to $NO_2^+$, $CH_2O_2^+$ may be useful for identifying ambient aqSOA formed through droplet processing.

Our results demonstrate that soot aerosols in Fresno are enriched with FOOA$_{BC}$ and inorganic ammonium nitrate during the winter fog event. Both components are hygroscopic, and in conjunction with the larger aerosol size distribution,

suggest that soot particles that have undergone cloud or fog processing may serve as effective cloud condensation nuclei. This finding has important implications for accurately modelling the removal rate of black carbon from the atmosphere in global climate models. Furthermore, we observed a correlation between the BC absorption enhancement (E$_{abs}$) and FOOA$_{BC}$ fraction. Further research is needed to determine whether this relationship is primarily influenced by the coating composition, or if it is influenced by covariation with aerosol morphology and size.

**Competing Interests**

At least one of the (co-)authors is a member of the editorial board of Atmospheric Chemistry and Physics

**Acknowledgments**

This work was funded by the U.S. Department of Energy (DOE) Atmospheric System Research (ASR) program (DE-SC0022140 and DE-SC0020182), and California Air Resources Board (CARB) (agreement 13-330). R.F. acknowledges

funding from the Jastro-Shields Research Award, Donald G. Crosby Fellowship and George Alexeeff Memorial Fellowship from the University of California at Davis

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





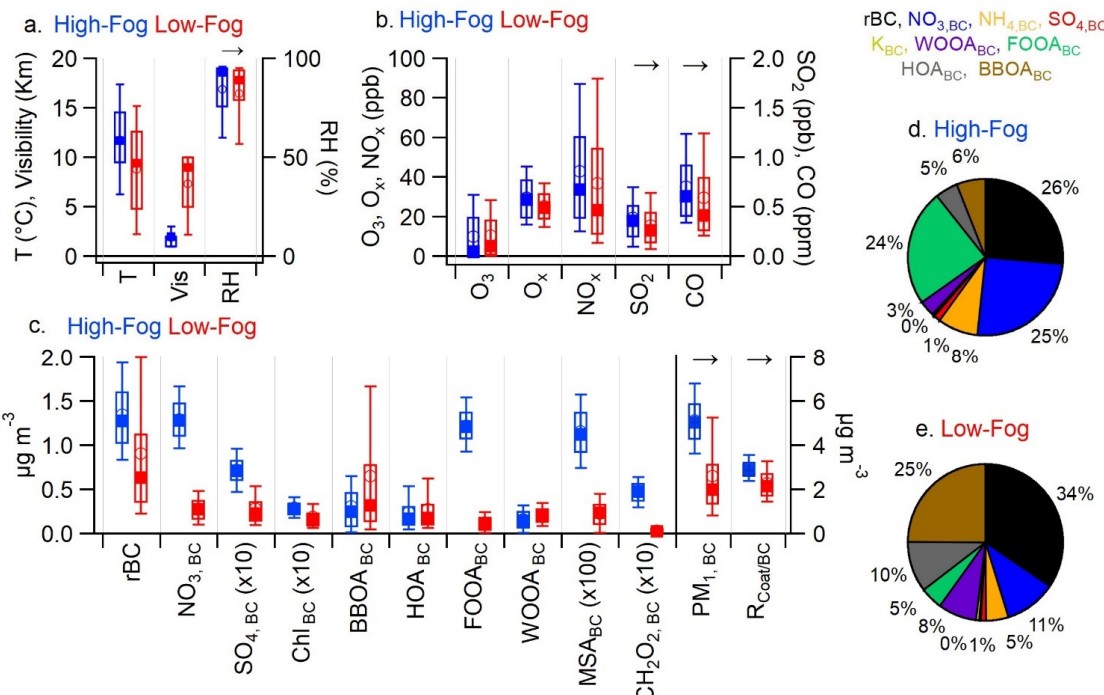

**Figure 1: Comparison of (a) meteorological variables, (b) gas phase compounds and (c) aerosol phase species between the high-fog (blue symbols) and low-fog (red symbols) periods.** The solid and open markers indicate the median and mean respectively, the box indicates the 25th-75th percentile, and whiskers indicate 10th-90th percentiles. $R_{Coat/BC}$ is the dimensionless ratio between the coating mass and rBC mass. **(d)** Average soot particle composition during the high-fog period. **(e)** Average soot particle composition during the low-fog period.





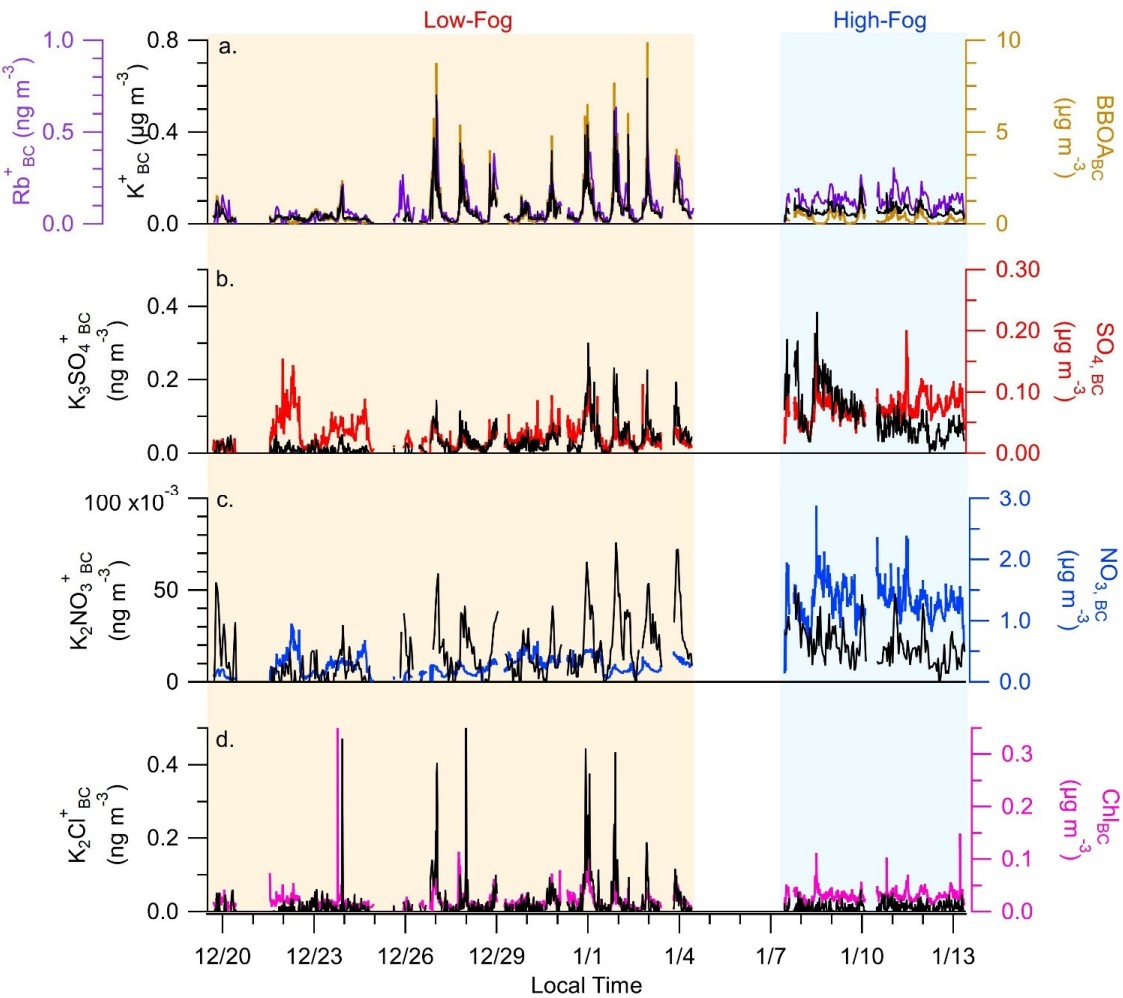

**Figure 2: Temporal variation of (a) $K^+_{BC}$, $Rb^+_{BC}$ and $BBOA_{BC}$, (b) $K_3SO_4^+$ and $SO_{4,BC}$, (c) $K_2NO_3^+$ and $NO_{3,BC}$ and (d) $K_2Cl^+$ and $Chl_{BC}$.**



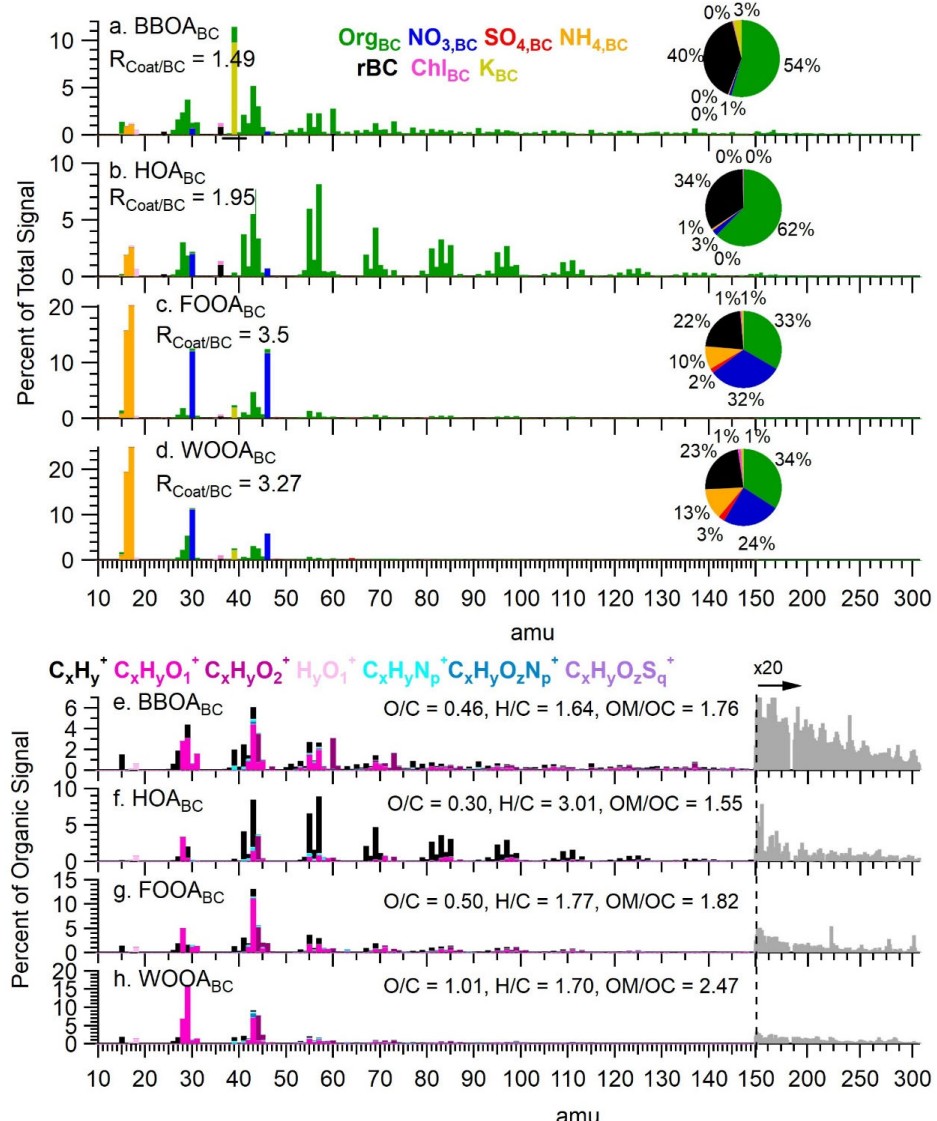

Figure 3: (a-d) Mass spectra of rBC containing PMF factors colored by species. y-axis is percent of total nitrate equivalent signal.
Pie chart insets depict the mass fraction of each species.  (e-h) Organic HRMS Spectra of the PMF factors colored by ion family.
Unit mass resolution data was used at values greater than 150 amu, and are scaled by 20 for clarity.





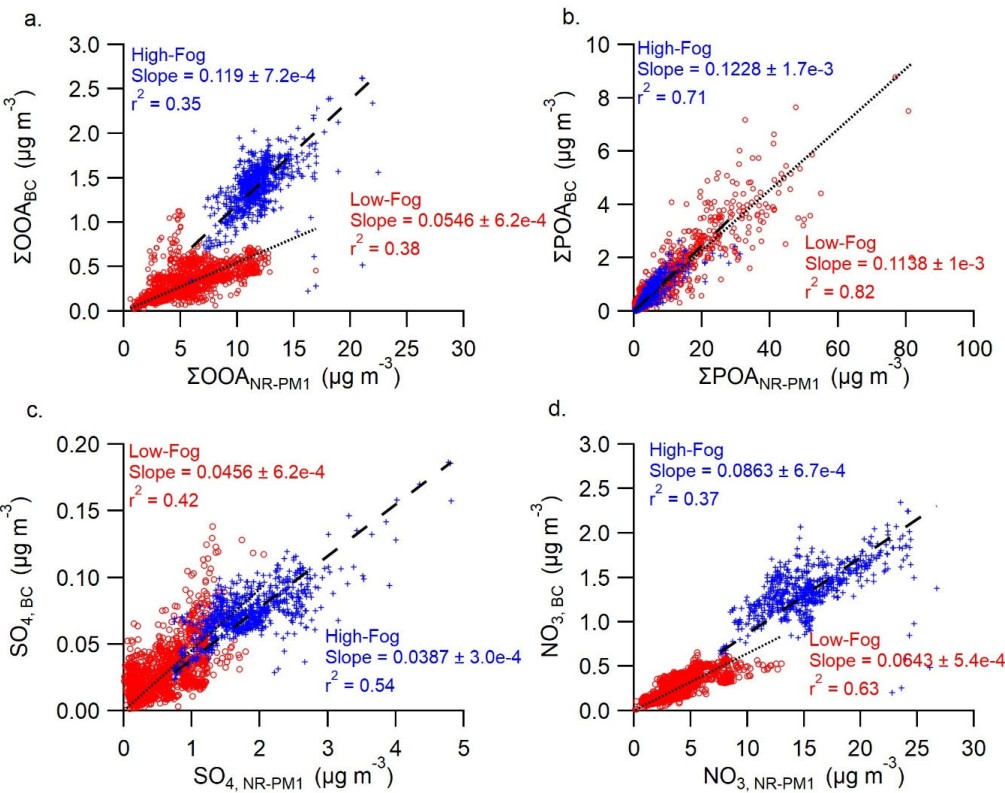

**Figure 4: Correlation between the (a) sum of OOA factors, (b) sum of POA factors, (c) sulfate and (d) nitrate from SP-AMS and HR-AMS. Data is separated between the low-fog period (Red circles) and high-fog period (blue crosses). The orthogonal distance regressions forced through the origin are also shown for the low-fog (dotted line) and high-fog periods (dashed lines) with fitting parameters included on the figure.**




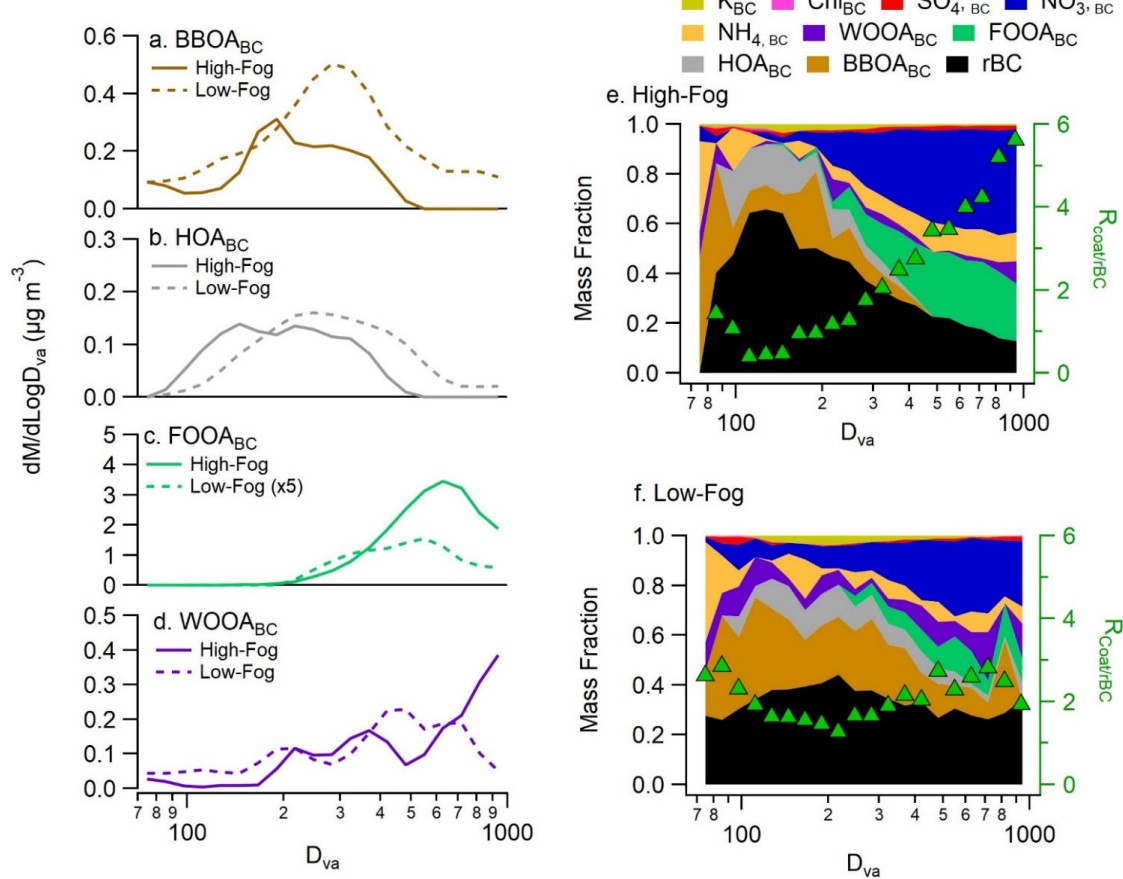

**Figure 5: (a-d)** Average size distributions of soot particle factors during both the high-fog period and low-fog period. Average aerosol composition as a function of size for the **(e)** high-fog period and **(f)** low-fog period. Size dependent coating thickness as a function of size is shown in the green triangles.



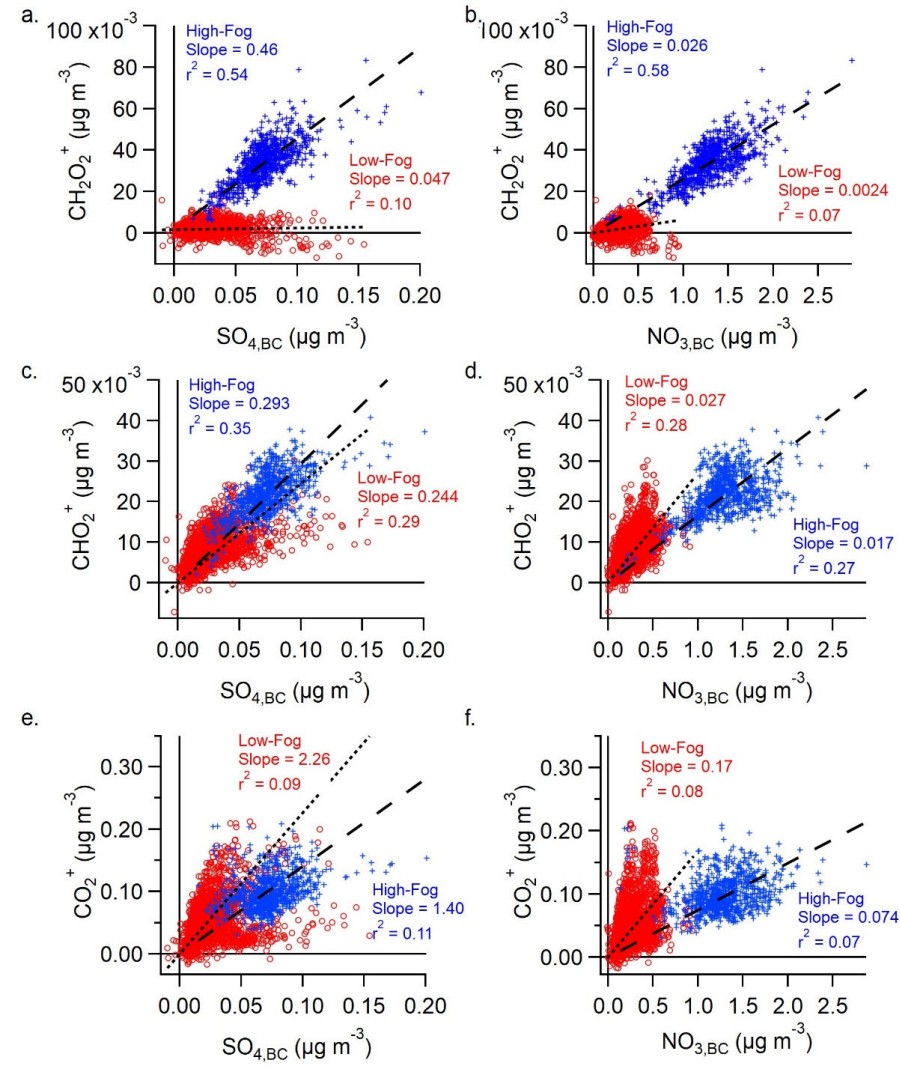

**Figure 6: Correlations between carboxylic acid tracers and (a,c,e) sulfate or (b,d,f) nitrate. Data is separated between the low-fog period (Red circles, dotted line) and high-fog period (blue crosses, dashed line). Trend lines are orthogonal distance regressions and forced through the origin.**





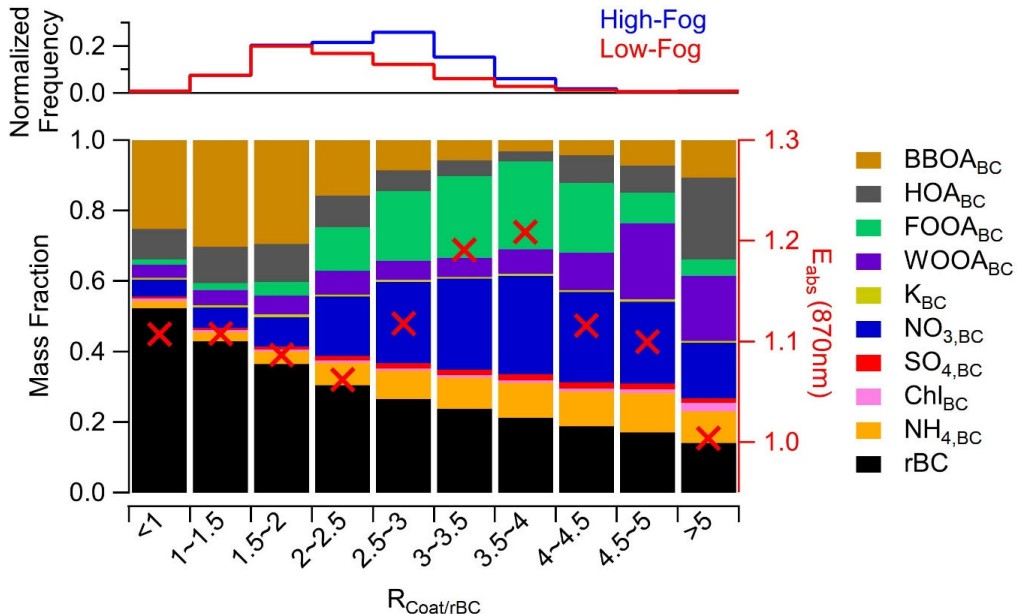


**Figure 7: Fractional mass contribution of aerosol species binned by rBC coating thickness. Red crosses indicate the average absorption enhancement for each bin. The top panel shows the normalized frequency of the occurrence of different coating thicknesses, separated between the two periods.**





**Table 1: Comparison of average (± 1SD) concentration of relevant organic tracer ions during the high-fog period and low-fog period.**

| Fragment Ion | High-Fog (ng m$^{-3}$) | Low-Fog (ng m$^{-3}$) | Notes | Reference |
|---|---|---|---|---|
| $CO_2^+$ | 189 ± 57.4 | 100. ± 75.9 | Organic acid tracer | Duplissy et al., 2011 |
| $CHO_2^+$ | 30.9 ± 7.37 | 11.9 ± 6.53 | Organic acid tracer | Jiang et al., 2021 |
| $CH_2O_2^+$ | 47.5 ± 14.2 | 3.06 ± 4.34 | Fog processing tracer | This work |
| $CH_3SO_2^+$ | 1.37 ± 0.427 | 0.277 ± 0.273 | MSA tracer | Ge et al., 2012a |
| $C_2H_4O_2^+$ | 46.1 ± 27.1 | 51.6 ± 66.0 | BBOA tracer | Cubison et al., 2011 |
| $C_4H_9^+$ | 74.6 ± 39.8 | 50.6 ± 54.7 | HOA tracer | Zhang et al., 2005 |