# Peer review of "Source Apportionment of Soot Particles and Aqueous-Phase Processing of Black Carbon Coatings in an Urban Environment"

_EGUsphere, 2023_

## Author Comment (AC1)

**Response to reviewers**

We thank the reviewers for their thoughtful and valuable comments, and we have incorporated their suggestions into the revised manuscript. Listed below are our point-to-point responses to the comments (in italic) and the corresponding manuscript revisions in quotation marks.

**Reviewer 1**

*General Comments:*

*This manuscript is well written and the results are nicely structured and discussed in great details. The authors present a field investigation of the effect of fog conditions on the chemical and physical properties of soot-containing particles through secondary processing during winter in the San Joaquin Valley. Enrichment in oxygenated organic aerosols and ammonium nitrate on BC-coating were observed during the fog event. The fog-related oxidized organic aerosols, are produced mainly by relatively fast aqueous phase processing of BBOA within fog droplets. The use of the potassium to BC ratio to estimate the fraction of BB contributing to the secondary formed $OOA_{BC}$ was a very interesting approach.*
*The comparison with the colocated AMS ensemble measurements provide valuable insights on the mixing state of the $PM_1$, with high-fog conditions favoring internally mixed BC with secondary OA and ammonium nitrate. Furthermore, they evidence a potential link between soot particle coated with secondary material from frog processing and changes in light absorption properties. The coating of BC by SOA and ammonium nitrate also lead to more hygroscopic soot particles and therefore more effective CCN.*
*I recommend this manuscript for publication after addressing minor comments. In particular, the authors caution about potential bias on the use of the $CH_2O_2^+$ fragment as tracer for aqueous processing. The influence of intense $NO_2^+$ signal, especially during the fog event when $NO_3$ concentrations drastically increased, which may bias high the intensity of $CH_2O_2^+$ (see comments).*

*Minor Comments:*
1. *Page 4 line 129: "the laser vaporizer RIE is varies from this", remove "is" and maybe add at least the RIE used in the SI. Is it 0.05 for rBC based on (Collier et al. 2018)?*

The typo was corrected.  Additionally, the sentence in question was rephrased to clarify the potassium RIE that was used.  The line now reads:

> "An RIE of 2.9 was used for potassium quantification (Drewnick et al., 2006) while rubidium and potassium-containing salts are presented in nitrate equivalent concentration (i.e. RIE of 1).  However, the previously established potassium RIE of 2.9 was determined using a ToF-AMS equipped with a thermal vaporizer and it is possible that the RIE may differ when the laser vaporizer is utilized."

2. *Page 7 lines 219-221: "In contrast, $K_3SO_4^+$ concentrations were elevated during the high-fog period compared to the low-fog period. The correlations between $K_3SO_4^+$ and $SO_{4,BC}$ concentrations showed variable slopes, gradually decreasing over the course of the high-fog period (Fig. 2b)." Are those different slopes linked to various source of $SO_{4,BC}$ or $K_3SO_4^+$? And why are $K_3SO_4^+$ concentrations high at the beginning of the fog period (following $NO_{3,BC}$) and then decrease whereas BBOA concentrations are low and FOOA ones (Figure S4) remain stable?*

The variations in slopes can be attributed to the distinct formation mechanisms associated with $SO_{4,BC}$ and $K_2SO_4$. Specifically, $K_2SO_4$ is expected to form through acid replacement reactions involving KCl, while sulfate forms due to $SO_2$ oxidation. The moderate correlation observed between $K_3SO_4^+$ and $K^+$ (and BBOA) supports the connection of $K_2SO_4$ with biomass burning. In addition, we observed a moderate correlation between $K_3SO_4^+$ and $SO_2$, both of which exhibited a decreasing trend during the high fog period. In contrast, the concentration of $SO_{4,BC}$ remained relatively stable, likely a result of the balance between enhanced aqueous-phase formation and wet deposition of sulfate, facilitated by fog events. Although the specific drivers of the changes in $K_3SO_4^+$ concentration in this work are not clearly known, the revised manuscript now provides more details, including the addition of the concentration of $SO_2$ to Fig. 2b.

3.   *Page 13 line 411-413 and 425-428: Although the $CH_2O_2^+$ signal appears real based on Figure S13a, its intensity is much lower than the $NO_2^+$ fragment, which could bias high the intensity of $CH_2O_2^+$ signal. This is even more relevant during the high-fog period, as $NO_3$ concentrations increase and so does $NO_2^+$ and $CH_2O_2^+$. If that's the case, their temporal variations would match (as shown in Figure S6b) and as a result, they will be grouped into a same factor by PMF. You may want to show the m/z 46 HR fitting peak averaged over the low and high-fog periods to support your point. I appreciate the caution about the peak fitting in the conclusion (page 16 line 500-502), but it might be better to address it earlier in the discussion.*

As the reviewer stated, it is possible that the $CH_2O_2^+$ signal is influenced by the neighboring $NO_2^+$ peak, however, it is clear that not all of the temporal variation in $CH_2O_2^+$ can be explained by variation in $NO_2^+$. This can be seen by the variable slope and moderate $r^2$ in Fig 6b. Following the reviewer's recommendation, we have added a direct comparison of the HR fitting for a portion of the low-fog period to Fig S13 alongside the HR fitting for the high-fog period as well as a comparison of the timeseries of $CH_2O_2^+$ and $NO_2^+$ and the corresponding scatterplot. The updated figure is shown below and additional text was added to page 14 to describe the new figure.

[Figure]

**Figure S13: (a) SP-AMS mass spectrum of pure oxalic acid sampled in the laboratory. (b) Reference mass spectrum for oxalic acid from the NIST databases. (c) Time series of $CH_2O_2^+$ and $NO_2^+$. (d) Scatterplot of $CH_2O_2^+$ and $NO_2^+$ signal. High**

**resolution peak fitting of m/z 46 during a representative section of the (e) high-fog period and (f) low-fog period. Note difference in y-axis scale.**

4. *What is the contribution of BC-containing particles to PM$_1$ fraction between low and high-fog periods? Could be added to Fig 1.*

A line was added to state the change in BC-containing fraction to total PM$_1$. The following sentence was added to page 6:

> "This increase was also associated with an elevated fraction of PM$_1$ associated with rBC, increasing from 15% during the low -period to 18% during the high-fog period."

*Supplementary information:*

1. *S1.1: What was the RH after the dryer, as it may affect the optical measurements.*

Added a clarification that the RH for the dried aerosol prior to optical measurements was <20% as stated in (Cappa et al., 2019).

2. *S1.3 lines 85-90: In the laser mode vaporization of the SP-AMS, $f_{CO2}$ can result from non-refractory $CO_2^+$ of the organic coating and refractory $CO_2^+$ from BC thermal decomposition, do you think it could result in an overestimation of $\kappa_{Org}$?*

Previous work (i.e. Corbin et al., 2014) has found enhanced signal at *m/z* 44 during vaporization of rBC. The line has been updated accordingly:

> "Signal at *m/z* 44 can also be produced through the decomposition of oxygenated functional groups on the BC surface (Corbin et al., 2014), resulting in a potential overestimation of $\kappa_{org}$."

3. *Fig S8: if available, the diurnal variations of T, RH and wind speed/direction during low and high-fog period could be useful information to add in this figure.*

The diurnal profiles of temperature, relative humidity, windspeed and wind direction have been added to the figure. Updated version of Fig. S8 is shown below. The text has been updated accordingly.

[Figure]

4. *Fig S11: "Top panels show the scaled residual between the measured and modeled size distributions." Is there something missing in the figure?*

The typo in the figure caption has been removed.

**Reviewer 2**

*This manuscript describes the chemical composition of BC aerosols, particle optical properties, sources of soot particles, and atmospheric processes affecting the properties of BC coatings. Studies have found that aqueous-phase reactions facilitated by fog droplets had a significant impact on the thickness and chemical composition of BC coatings, which represents interesting and meaningful academic research. I recommend the publication of the manuscript after the following points have been addressed.*

*I only have several minor questions about the manuscript.*

*Point 1: Lines 172-173: "These observations provide clear evidence that the presence of fog droplets promoted the formation of nitrate on BC particles", is it also possible that the presence of nitrate can cause high fog environments?*

The fact that nitrate particles are effective CCN does suggest that increased nitrate levels can promote fog formation. Notably, a recent study on the long-term frequency of fog events in the Central Valley of California has indicated that air pollution has been a significant factor in the longer-term temporal and spatial changes in fogs in the region (Gray et al., 2019). However, short-term fluctuations in fog, such as the occurrence of the high fog period observed in this study, are primarily governed by meteorological conditions.

*Point 2: Lines 194-195: The proportion of organic compounds in the high-log period is smaller than that in the low-log period, but the concentration of $Org_{BC}$ is higher in the high-log period. More details of the possible mechanisms should be discussed here.*

Although the absolute $Org_{BC}$ concentration increases in the high-fog period, the fractional contribution decreases due to the abundance of ammonium nitrate. The section was updated to:

> "Organic compounds were the most abundant species on soot aerosol, contributing 38% and 48% of $PM_{1,BC}$ mass during the high-fog period and low-fog period, respectively. However, the $Org_{BC}$ was significantly higher during the high-fog period, increasing from $1.25 \pm 1.13$ µg m$^{-3}$ to $1.96 \pm 0.57$ µg m$^{-3}$. The smaller $Org_{BC}$ mass fraction during the high fog period was primarily driven by the accumulation of $NO_{3,BC}$ and $NH_{4,BC}$. Additionally…"

*Point 3: There is no apparent logical relationship between the subsections, the article logic and analysis sequence need to be reorganized for a more natural transition between the subsections.*

We have made significant revisions in the results section to address this comment. The six subsections now center around the following topics:
3.1. Influence of Winter Fog Events on Soot Aerosol Composition and Properties in Fresno
3.2. Sources and Chemical Processing of Soot Particles in Fresno
3.3. Effect of Fog Events on the Partitioning of Aerosol Species between Soot and BC-Free Particles
3.4. Effect of Fog Processing on Soot Particle Size Distribution
3.5. Chemical Signatures of aqSOA Formation Observed on Soot Particles
3.6. Influence of Aqueous-Phase Processing on Soot Particle Absorption Properties and Hygroscopicity

*Point 4: The mention of "However, the diurnal profiles of HOA$_{BC}$ showed notable differences between the two periods" in Line 261 appears to have a hasty explanation. Evidence to prove specific differences in traffic patterns could make it more convincing, and the mentioning of the possible relationship between these differences and variations in boundary layer height seems somewhat abrupt.*

Further details were included regarding the differences in diurnal profiles of HOA$_{BC}$ between the two period.  The paragraph now reads:

> "The average organic concentrations of HOA$_{BC}$ during the low-fog and high-fog periods were relatively consistent at 0.30±0.32 µg m$^{-3}$ and 0.24±0.23 µg m$^{-3}$, respectively, indicating a stable emission of soot particles from vehicular sources throughout the campaign. The diurnal profile of HOA$_{BC}$ was similar to gas-phase NO$_x$,  however HOA$_{BC}$ showed notable differences between the two periods (Fig. S6, S7b). During the high-fog period, two diurnal peaks of similar magnitudes were observed.  The peak in the morning (09:00-10:00) corresponded to rush hour, while the peak in the evening (19:00-23:00) was influenced by a combination of rush hour, decreasing boundary layer height and other, late night combustion activity.  The peak during the late evening (22:00-23:00) occurred later than expected from rush hour, but was similar to previous observations of HOA in Fresno (Sun et al., 2022; Young et al., 2016).  In contrast, during the low-fog period, the morning rush-hour peak was nearly absent, and the evening peak occurred two hours earlier peaking between 20:00 – 22:00.  The differences in the diurnal profiles may be results of reduced commuter traffic during the winter holidays, or lower boundary layer height due to colder temperatures."

*Point 5: lines 423-424 "This finding is in contrast to measurements in China, where primary BB emissions accounted for 30% of the oxalate mass", however, it did not mention the differences or the specific details of the data which has "no significant relationship between CH$_2$O$_2^+$ and BBOA$_{BC}$ (r$^2$ = 0.02)".*

We hypothesize that primary BBOA emissions were not a major source of oxalate in this work based on the poor correlation of CH$_2$O$_2^+$ and the BBOA factor.  This sentence has been revised to improve clarity.  It now reads:

> "However, in this study, we see no correlation between CH$_2$O$_2^+$ and BBOA$_{BC}$ (r$^2$ = 0.02), suggesting that primary emission of oxalate from biomass burning activity is not a significant source of CH$_2$O$_2^+$. This finding is in contrast to measurements in China, where primary BB emissions accounted for 30% of the oxalate mass (Yang et al., 2014)."

References:

Cappa, C. D., Zhang, X., Russell, L. M., Collier, S., Lee, A. K. Y., Chen, C. L., Betha, R., Chen, S., Liu, J., Price, D. J., Sanchez, K. J., McMeeking, G. R., Williams, L. R., Onasch, T. B., Worsnop, D. R., Abbatt, J. and Zhang, Q.: Light Absorption by Ambient Black and Brown Carbon and its Dependence on Black Carbon Coating State for Two California, USA, Cities in Winter and Summer, J. Geophys. Res. Atmos., 124(3), 1550–1577, doi:10.1029/2018JD029501, 2019.

Corbin, J. C., Sierau, B., Gysel, M., Laborde, M., Keller, A., Kim, J., Petzold, A., Onasch, T. B., Lohmann, U. and Mensah, A. A.: Mass spectrometry of refractory black carbon particles from six sources: Carbon-cluster and oxygenated ions, Atmos. Chem. Phys., 14(5), 2591–2603, doi:10.5194/acp-14-2591-2014, 2014.

Drewnick, F., Hings, S. S., Curtius, J., Eerdekens, G. and Williams, J.: Measurement of fine particulate and gas-phase species during the New Year's fireworks 2005 in Mainz, Germany, Atmos. Environ., 40(23), 4316–4327, doi:10.1016/j.atmosenv.2006.03.040, 2006.

Gray, E., Gilardoni, S., Baldocchi, D., McDonald, B. C., Facchini, M. C. and Goldstein, A. H.: Impact of Air Pollution Controls on Radiation Fog Frequency in the Central Valley of California, J. Geophys. Res. Atmos., 124(11), 5889–5905, doi:10.1029/2018JD029419, 2019.

Sun, P., Farley, R. N., Li, L., Srivastava, D., Niedek, C. R., Li, J., Wang, N., Cappa, C. D., Pusede, S. E., Yu, Z., Croteau, P. and Zhang, Q.: PM2.5 composition and sources in the San Joaquin Valley of California : A long-term study using ToF-ACSM with the capture vaporizer, Environ. Pollut., 292(PA), 118254, doi:10.1016/j.envpol.2021.118254, 2022.

Yang, F., Gu, Z., Feng, J., Liu, X. and Yao, X.: Biogenic and anthropogenic sources of oxalate in PM 2 . 5 in a mega city , Shanghai, Atmos. Res., 138, 356–363, doi:10.1016/j.atmosres.2013.12.006, 2014.

Young, D. E., Kim, H., Parworth, C., Zhou, S., Zhang, X., Cappa, C. D., Seco, R., Kim, S. and Zhang, Q.: Influences of emission sources and meteorology on aerosol chemistry in a polluted urban environment: Results from DISCOVER-AQ California, Atmos. Chem. Phys., 16(8), 5427–5451, doi:10.5194/acp-16-5427-2016, 2016.